# Antibody Responses to mRNA COVID-19 Vaccine Among Healthcare Workers in Outpatient Clinics in Japan

**DOI:** 10.3390/vaccines13010090

**Published:** 2025-01-18

**Authors:** Teruhime Otoguro, Keita Wagatsuma, Toshiharu Hino, Yusuke Ichikawa, Tri Bayu Purnama, Yuyang Sun, Jiaming Li, Irina Chon, Hisami Watanabe, Reiko Saito

**Affiliations:** 1Infectious Diseases Research Center of Niigata University in Myanmar, Niigata University, Niigata 950-8510, Japan; hwatanabe@med.niigata-u.ac.jp (H.W.); jasmine@med.niigata-u.ac.jp (R.S.); 2Division of International Health (Public Health), Graduate School of Medical and Dental Sciences, Niigata University, Niigata 950-8510, Japan; waga@med.niigata-u.ac.jp (K.W.); n22c202a@mail.cc.niigata-u.ac.jp (Y.I.); tribayupurnama@uinsu.ac.id (T.B.P.); sunyuyang@med.niigata-u.ac.jp (Y.S.); lijiaming@med.niigata-u.ac.jp (J.L.); irinachon@med.niigata-u.ac.jp (I.C.); 3Institute for Research Administration, Niigata University, Niigata 950-8510, Japan; 4Hino Pediatric Internal Medicine Clinic, Nishinomiya 662-0927, Japan; thino@hi-ho.ne.jp; 5Faculty of Public Health, Universitas Islam Negeri Sumatera Utara, Medan 20371, Indonesia

**Keywords:** SARS-CoV-2, COVID-19, antibody, healthcare workers

## Abstract

Background: This study aimed to assess the antibody response to SARS-CoV-2 vaccines among healthcare workers (HCWs) from multiple outpatient clinics in Japan, examining the effects of baseline characteristics (e.g., sex, age, underlying condition, smoking history, occupation) and prior infections. Methods: A total of 101 HCWs provided serum at four time points between October 2020 and July 2023. HCWs received two to six doses of mRNA vaccine (BNT162b2 or mRNA-1273). Anti-nucleocapsid (N) and anti-spike (S) IgG antibodies against the ancestral Wuhan strain were measured using the Abbott Architect™ SARS-CoV-2 IgG assay. Univariate and regression analysis evaluated factors such as past infections, age, sex, smoking, underlying condition, and occupation. Results: After four to six doses, the median anti-S IgG titer in uninfected HCWs was 1807.30 BAU/mL, compared to 1899.89 BAU/mL in HCWs with prior infections. The median anti-N IgG titer was 0.10 index S/C in uninfected HCWs and 0.39 index S/C in infected HCWs. HCWs with prior infection had anti-S IgG titers 1.1 to 5.8 times higher than those without. Univariate and multivariate analyses indicated infection and vaccination significantly increased anti-S and anti-N IgG titers. Age, sex, smoking history and occupation did not influence antibody titers while underlying conditions were associated with lower anti-N IgG titers. Conclusions: Infection and vaccination were strongly associated with an increase in anti-S and anti-N IgG titers; however, the impact of hybrid immunity appeared to be limited and varied depending on the timing of the sampling. These findings provide valuable insights for developing personalized vaccination strategies and future vaccine development.

## 1. Introduction

The severe acute respiratory syndrome coronavirus 2 (SARS-CoV-2) has profoundly affected global public health since its emergence in late 2019, resulting in the novel coronavirus disease 2019 (COVID-19) pandemic [1]. Vaccination against SARS-CoV-2 has demonstrated effectiveness in mitigating the spread of the virus and controlling the COVID-19 pandemic [2]. By the end of 2020, multiple vaccine platforms, including mRNA, viral vector, protein subunit, and inactivated vaccines, had been introduced, and approximately 5.47 billion doses had been administered globally by December 2023 [2,3].

COVID-19 vaccination in Japan began in February 2021 with the administration of mRNA vaccine BNT162b2 (Pfizer-BioNTech, New York, USA) encoding a Wuhan-like spike to healthcare workers (HCWs), followed by the rollout to the elderly population starting in April 2021 [4]. In May 2021, the mRNA vaccine mRNA-1273 (Moderna Inc., Cambridge, MA, USA) was introduced at mass vaccination sites, and subsequently, the initial vaccination series was expanded to include all individuals aged 12 and older. By December 2023, 82% of the population had completed the primary COVID-19 vaccination series, while 69% had received at least one booster dose [2]. The third dose of the COVID-19 vaccine was introduced in December 2021, with the fourth dose administered starting in June 2022 to individuals aged 60 and older and those at high risk, followed by HCWs from July 2022 [4]. In September 2022, the bivalent BNT162b2 vaccine (Wuhan/Omicron BA.1) became available for booster doses, and in October 2022, the bivalent BNT162b2 vaccine (Wuhan/Omicron BA.4-5) was introduced, facilitating the rollout of fifth doses [4]. From May 2023, additional doses of the bivalent vaccine targeting the Omicron variant were provided to individuals aged 65 and older, those aged 5–64 with underlying medical conditions, and HCWs or staff at elderly care facilities [4]. As the Omicron XBB lineage became predominant, a monovalent mRNA BNT162b2 and mRNA-1273 vaccine (Omicron XBB.1.5) was developed and introduced in September 2023; the fall 2023 vaccination campaign began, offering this vaccine to all individuals aged six months and older [4]. The overall immune response to BNT162b2 and mRNA-1273 has been reported [5,6], inducing SARS-CoV-2 specific spike-protein (and/or its RBD) B-cells and neutralizing the antibody response and generation of specific polyfunctional CD8+ and CD4+ T-cells. Nevertheless, the emergence of SARS-CoV-2 variants, particularly Omicron, has raised concerns regarding their ability to partially evade immune responses elicited by prior infections or vaccinations, which could potentially compromise immunity and elevate the risk of breakthrough infections [7]. Empirical studies have demonstrated that vaccine efficacy can be modulated by factors such as prior infection history, vaccination status, individual characteristics, pre-existing health conditions, and occupational exposure [8,9]. HCWs, in particular, face a heightened risk of SARS-CoV-2 infection due to their continuous exposure to infected individuals and contaminated environments. For HCWs frequently exposed to high viral loads, understanding the kinetics of antibody production and persistence is crucial for safeguarding individual and community health [10,11].

Previous Japanese HCW studies have mainly focused on antibody responses in large hospital settings, but there is a lack of data on antibody levels in outpatient clinics throughout Japan [12,13]. Due to differences in patient characteristics and infection control measures, the infection risk among HCWs may differ between large hospitals, where the number of HCWs is higher and the number of SARS-CoV-2 infected patients in severe conditions is greater and outpatient clinic settings, where there are fewer HCWs and the cases being consulted are fewer and milder cases.

Consequently, we investigated antibody levels across 19 Japanese outpatient clinics, instead of focusing on hospital settings. This study sought to elucidate the impact of various factors, including age, sex, occupation, preexisting conditions, and vaccination history, on variations of antibody levels among HCWs in outpatient clinics. Additionally, we examined the impact of infection and vaccination histories on antibody titers and their association. By understanding the determinants of antibody levels, we can more effectively protect high-risk groups, such as HCWs, and enhance public health interventions to interrupt COVID-19 transmission.

## 2. Materials and Methods

### 2.1. Study Design and Participants

In this study, we invited participants using the mailing list of 34 outpatient clinics affiliated with the research group Kinki Ambulatory Pediatrics Study Group (KAPSG) and successfully enrolled 136 participants from 19 outpatient clinics across nine different prefectures in Japan, namely Fukui, Nara, Hyogo, Osaka, Kyoto, Tokyo, Aichi, Kagawa, and Fukuoka, between October 2020 and August 2023 (Figure 1A). Only outpatient clinics likely to cooperate with the study were approached, and all recruited participants ultimately took part, resulting in a 100% participation rate. However, 33 participants were excluded due to resignations or voluntary withdrawals during the study period. Informed consent was obtained from the participants, along with demographic and clinical data, such as sex, age, underlying condition (i.e., hypertension, heart disease, cancer, photosensitivity, glaucoma, dyslipidemia, hyperlipidemia, sleep apnea syndrome), smoking history (yes or no), occupation (i.e., doctor, nurse, receptionist or childminder) at the first sampling before the COVID-19 vaccination started. Participants recorded their COVID-19 vaccination (i.e., none, one dose, two doses, three doses, four doses, five doses, six doses) on a survey at each blood sampling, noting the year and month they received each vaccine dose. They also documented whether they had contracted COVID-19 within the past six months at each blood sampling (yes or no). During the study period, the Japanese government conducted stringent surveys of COVID-19 cases. Therefore, participants were required to confirm any symptoms such as fever, cough, runny nose, or fatigue using a rapid diagnostic test kit or reverse-transcription PCR (RT-PCR) test to confirm SARS-Co-V2 infections. Medical doctors or nurses who participated in this study collected blood samples from the forearm vein using a winged needle (21G) and EDTA-2Na/F-treated vacuum blood collection tubes (5 mL), and plasma samples were centrifuged before measurement and stored in a freezer at –20 °C until testing. HCWs received a monovalent vaccine of BNT16b2 (Wuhan) for the first to fourth dose, a bivalent vaccine of BNT162b2 (Wuhan/BA.1, Wuhan/BA.4-5) for the fourth to sixth dose, a monovalent vaccine of BNT16b2 (XBB.1.5) for the sixth dose and a bivalent vaccine of mRNA-1273 (Wuhan/BA.4-5) for the fourth and sixth dose according to the vaccination schedules recommended by the Japanese government. This study was approved by the Ethics Committee of the Society of Ambulatory and General Pediatrics of Japan on 7 April 2020 (No. 2020-4).

### 2.2. National Genomic Epidemiology Data from Nextstrain

To support the analysis of the antibody, timelines of the genomic epidemiology of SARS-CoV-2 in Japan were used from the data in Nextstrain (https://nextstrain.org/ncov/gisaid/global/all-time?f_country=Japan, accessed on 24 September 2024). The dataset was filtered to include genomes sampled in Japan, resulting in 49 of 3946 genomes being collected between January 2020 and September 2024.

### 2.3. Measurement of Anti-S and -N Antibodies Titer

The anti-spike (S) and anti-nucleocapsid (N) SARS-CoV-2 IgG antibodies (Abs) against the ancestral Wuhan strain were measured using the Chemiluminescent Microparticle Immunoassay (CMIA) technique by SARS-CoV-2 IgG Reagent Kit (Cat. No. 6R86; Abbott, Chicago, IL, USA) for anti-N IgG Ab and AdviseDx SARS-CoV-2 IgG II Reagent Kit (Cat. No.06S61; Abbott) for anti-S IgG Ab [14,15,16,17,18]. The manufacturer’s recommended positive cut-off index was ≥1.4 (index Sample/Control [S/C]) for anti-N antibodies and ≥50.0 AU/mL for anti-S antibodies for the Abbott [17,18]. HCWs whose anti-S IgG Ab titer exceeded the upper limit (40,000 AU/mL) at any point were treated as having the upper limit titer. These tests were performed using the high-throughput ARCHITECT i2000SR (Abbott) [19]. SARS-CoV-2 binding antibody units per mL (BAU/mL) were calculated according to the WHO international standard for anti-SARS-CoV-2 immunoglobulin using conversion factors (Abbott 0.142) [20].

### 2.4. Statistical Analysis

No statistical sample size calculations were conducted in this study because, at the start of the research, the data required for estimation were not available. Additionally, the limited existing information on the targeted antibody responses and predictors, together with a lack of prior knowledge on expected effect sizes and distributions, made it challenging to perform a statistically accurate estimation. Therefore, considering feasibility, we opted to include all eligible HCWs who could be observed.

Descriptive statistics were performed with percentages or medians with interquartile ranges (IQRs). The univariate analysis compared groups between anti-S and N IgG Ab titers using a Mann–Whitney U test. To determine factors influencing anti-S and anti-N IgG antibody titers, this study employed a generalized linear regression model with a gamma distribution and a natural logarithmic link function with robust error variances [21]. Specifically, multivariable models were formulated for the anti-S and anti-N IgG Ab titers as response variables after adjusting for the following potential confounders: sex (male or female), age (<65 or ≥65), underlying condition (yes or no), occupation (doctor, nurse, receptionist, or others), number of vaccinations (none, two doses, three doses, or four to six doses) and previous SARS-CoV-2 infection (yes or no), and as fixed effects at the participant level. Due to the limited number of participants who received five or six vaccine doses, the data for these groups were combined to four to six doses. HCWs whose anti-S IgG Ab titer exceeded the upper limit at any point were treated as having the upper limit titer. The maximum likelihood method was used for inference, and the estimates, including adjusted regression *β* coefficients, 95% confidence intervals (CIs), and *p*-values were calculated. Statistical significance was set at a *p*-value of <0.05 (type I error) on a two-tailed test. All statistical analyses were performed using STATA version 15.1 statistical software (Stata Corp, College Station, TX, USA).

## 3. Results

### 3.1. Clinical Characteristics of Participants

A total of 136 HCWs were enrolled in this study between October 2020 and July 2023; eventually, 101 HCWs were eligible for the analysis. The remaining 35 HCWs were excluded from the analysis for the reason described in Figure 1A. The 101 HCWs in this analysis provided serum at each of the four time points (Figure 1B). The first blood sampling was implemented in the median 18 days (IQR: 12.0–24.0 days) before the vaccination started; the second sampling was implemented in the median 168 days (IQR: 161–175 days) after one dose, which is in the median 146 days (IQR: 140–153 days) after two doses; the third sampling was done in the median 104 days (IQR: 95–108 days) after three doses; the last fourth sampling in the median 319 days (IQR: 263–342 days) after four doses, which was in the median 199 days (IQR: 174–217 days) after five doses, and in the median 37 days (IQR: 28–39 days) after six doses, respectively. Table 1 shows the baseline characteristics of the eligible HCWs. The median age was 54.0 (IQR: 46.3–60.3) and most HCWs were female (83.2%) during the first blood sampling before vaccination. All HCWs received one- and two-dose vaccinations. Additionally, 99.0% received three doses, 86.1% received four doses, 63.4% received five doses, and 39.6% received six doses. The monovalent BNT162b2 (Wuhan) vaccine was used for the first dose (n = 101, 100%), second dose (n = 101, 100%), and third dose (n = 100, 99.0%), while the monovalent BNT162b2 (Wuhan; n = 43, 42.3%), bivalent BNT162b2 (Wuhan/BA.1; n = 8, 7.9%), bivalent BNT162b2 (Wuhan/BA.4-5; n = 17, 16.8%), BNT162b2 (Unknown; n = 18, 17.8%), and mRNA-1273 (Unknown; n = 1, 1.0%) were used for the fourth dose. For the fifth dose, bivalent BNT162b2 (Wuhan/BA.4-5; n = 64, 63.4%) was used, and for the sixth dose, bivalent BNT162b2 (Wuhan/BA.1; n = 2, 2.0%), bivalent BNT162b2 (Wuhan/BA.4-5; n = 21, 20.8%), monovalent BNT162b2 (XBB.1.5; n = 11, 10.9%), BNT162b2 (Unknown; n = 1, 1.0%), bivalent mRNA-1273 (Wuhan/BA.4-5; n = 1, 1.0%), and unknown (n = 4, 4.0%) were used. During the study period, a total of 50 participants reported contracting COVID-19. Specifically, one HCW (1.0%) reported new infections within six months before the first blood sampling, one HCW (1.0%) before the second sampling, 11 HCWs (10.9%) before the third sampling, and 37 HCWs (36.6%) before the fourth sampling. The largest group of participants was receptionists (39.6%), followed by nurses (35.6%), doctors (19.8%), childminders (1.0%), and others (4.0%). A total of 30 (29.7%) HCWs had underlying conditions, including 15 with hypertension (14.9%), 2 with heart disease (2.0%), 2 with cancer (2.0%), and 4 others (4.0%; anemia, hay fever, sun allergy, osteoporosis, dyslipidemia, hyperlipidemia, glaucoma, sleep apnea). Seven (6.9%) HCWs reported having a smoking habit.

### 3.2. The Temporal Dynamics of Anti-S and -N Antibodies Titer

To evaluate the temporal change in antibody titers pre- and post-vaccination, we analyzed the anti-S and anti-N IgG antibody titers at the four stages of blood collection (Figure 2 and Table 2). Among all HCWs who participated in the study, the anti-S IgG levels were low at 0.36 BAU/mL at the first sampling before vaccination, from October 2020 to April 2021. However, after receiving the primary series of two vaccine doses, the levels increased to 163.7 AU/mL (455-fold increase) at the second sampling in October 2021. Following three doses, the levels rose to 1165.5 BAU/mL (3238-fold increase) at the third sampling from April to May 2022 to May 2022; subsequent booster doses (four to six doses) significantly increased, reaching 1808.5 BAU/mL (5024-fold increase) at the fourth sampling during June and July 2023. Overall, the anti-S IgG levels increased with each additional vaccine dose. Similarly, the seroprevalence of anti-S antibodies showed a marked increase, from 3.0% at pre-vaccination to 99.0% after one to two doses, 100% after three doses, and 100% after four to six doses (Table 2). On the other hand, anti-N displayed a modest low increase throughout the blood sampling period, at the first sampling before vaccination the median levels being 0.03 index S/C; at the second sampling, 0.03 index S/C (1.3-fold); at the third sampling, 0.04 index S/C (1.3-fold); at the fourth sampling, the levels remained low at 0.18 index S/C (6.0-fold). As such, the seroprevalence of anti-N Ab displayed a low level of 1.2%, 0.0%, 5.9%, and 7.9% at the corresponding four time points (Table 2). These results are derived from the relatively low infections of COVID-19 in the studied HCWs during the study period.

### 3.3. Comparison of Anti-S and -N Antibodies Titer by Infection History in Univariate Analysis

When comparing the anti-S IgG levels between those infected with SARS-CoV-2 and those without, the antibody levels were consistently higher in the HCWs with previous infections across all four blood sampling points. The anti-S IgG antibody titers were 1.1 to 5.8 times higher than those without a prior SARS-CoV-2 infection, particularly at the second sampling after one to two vaccine doses (median anti-S IgG; 162.4 BAU/mL versus 617.5 BAU/mL), at the third sampling after three vaccine doses (median anti-S IgG; 987.27 BAU/mL versus 5680.00 BAU/mL, *p* < 0.001), and at the fourth sampling after four to six vaccine doses (median anti-S IgG; 1807.30 BAU/mL versus 1899.89 BAU/mL) (Table 3 and Figure 3A). A significant difference was observed only at the third sampling after three vaccine doses, suggesting the potential influence of hybrid immunity. However, when considering the results as a whole, the impact of hybrid immunity appeared to be limited, as the increase in anti-S IgG antibody after four to six doses was moderate regardless of prior SARS-CoV-2 infection.

Notably, the anti-N IgG antibody titers were 3.9 to 48.3 times higher than those without a prior SARS-CoV-2 infection, particularly at the second sampling after one to two vaccine doses (median anti-N IgG; 0.03 index S/C versus 1.39 index S/C), at the third sampling after three vaccine doses (median anti-N IgG; 0.04 index S/C versus 1.93 index S/C, *p* < 0.001), and at the fourth sampling after four to six vaccine doses (median anti-N IgG; 0.10 index S/C versus 0.39 index S/C, *p* < 0.001) (Table 3 and Figure 3A). After three doses and four to six doses, the anti-N IgG Ab titer in HCWs with a previous SARS-CoV-2 infection significantly increased compared to their counterparts without an earlier SARS-CoV-2 infection (Table 3 and Figure 3B).

### 3.4. Comparison of Anti-S and -N Antibodies Titer in Various Factors Using Univariate Analysis

We further examined whether antibody titers varied across different factors at four time points using univariate analysis (Figure 4). Among HCWs without a previous SARS-CoV-2 infection, a temporal increase in anti-S IgG Ab titers was observed in both males and females, with no significant differences between sexes (Figure 4A). Similarly, no significant sex differences were found in anti-N IgG Ab titer at any time points (Figure 4B). Anti-S IgG Ab titers were slightly higher in HCWs aged 65 years or older compared to those younger than 65 at all time points (Figure 4C). However, anti-N IgG Ab in HCWs aged 65 years or older were slightly lower than those in younger HCWs (Figure 4D). At all time points, no statistical significances in anti-S IgG Ab titers were observed between HCWs with and without underlying conditions (Figure 4E). Conversely, anti-N IgG Ab titers were slightly lower in HCWs with underlying conditions compared to those without underlying conditions (Figure 4F). At all time points, no statistically significant differences in anti-S IgG Ab titers were observed between HCWs with and without a smoking history (Figure 4G). Similarly, anti-N IgG Ab titers showed no differences between those with and without a smoking history (Figure 4H). Anti-S IgG Ab titers were highest among doctors, followed by similar levels among nurses, receptionists and others, but there were no significant differences in anti-S IgG Ab titers between the occupational groups (Figure 4I). The anti-N IgG Ab titer was low across all occupational groups (Figure 4J).

In HCWs with a previous SARS-CoV-2 infection, no significant differences in either anti-S or anti-N Ab titers were observed across factors based on sex (Figure 4A,B), age (<65 or ≥65; Figure 4C,D), the presence of an underlying condition (Figure 4E,F), smoking history (Figure 4G,H), and occupation (Figure 4I,J). However, compared to HCWs without a previous SARS-CoV-2 infection, anti-S and anti-N IgG Ab titers tended to be higher across all factors.

### 3.5. Analysis of Anti-S and -N Antibodies Titer Using Multivariate Regression Model

To predict the factors influencing anti-S and anti-N IgG Ab titers, we utilized a multivariate generalized linear regression model (Table 4 and Table 5, respectively). The analysis indicated that sex, age, underlying condition, and occupation were not significantly associated with anti-S IgG Ab titers. However, a previous SARS-CoV-2 infection was strongly associated with increased anti-S IgG titers (*β* coefficient: 1.83, 95% CI: 0.71 to 2.95, *p*-Value < 0.01). Moreover, as the vaccination status progressed to two, three, and four to six doses, anti-S Ab titers demonstrated a corresponding 5.55-, 7.05-, and 7.40-fold increase relative to the non-vaccinated cohort (Table 4).

Regarding anti-N Ab titers, neither sex, age nor occupation showed significant associations. Anti-N Ab titers were significantly elevated in HCWs with a previous SARS-CoV-2 infection (*β* coefficient: 2.17, 95% CI: 1.75 to 2.59, *p*-Value < 0.001), whereas titers decreased in relation to the number of vaccinations, exhibiting a 0.12-, 0.73-, and 1.16-fold reduction compared to non-vaccinated HCWs (Table 5). Additionally, underlying condition were inversely associated with anti-N IgG titers (*β* coefficient: −0.38, 95% CI: −0.68 to −0.08, *p*-Value < 0.05).

## 4. Discussion

We explored the antibody responses to the mRNA COVID-19 vaccine and their predictors among HCWs in outpatient clinics in Japan from October 2020 to August 2023. Our data showed the anti-S IgG Ab titers significantly increased over time in HCWs. Additionally, the anti-S IgG titers in infected HCWs were higher than those without only after three doses of vaccination, suggesting the effect of hybrid immunity was limited. Both univariate analysis and log-gamma generalized linear model analysis showed that anti-S and N Ab titers increased based on previous infection and the number of SARS-CoV-2 vaccinations received by HCWs vaccinated with the mRNA vaccine. By contrast, factors such as age, sex, smoking history, and occupation did not influence anti-S and N Ab titers, whereas underlying conditions were associated with lower anti-N Ab titers.

Our results suggest that the impact of hybrid immunity appeared to be limited and varied depending on the timing of the sampling (Figure 3A). Increasing evidence supports that hybrid immunity is more effective than immunity from vaccination alone or infection alone. Goldberg et al. reported from Israeli data that immunity to SARS-CoV-2 infection and the BNT162b2 vaccine decreases over time, but individuals with prior infection maintain higher protection against infections than those who have only been vaccinated [22]. However, a study conducted in Peru found that a previous SARS-CoV-2 infection did not significantly affect the anti-S titer, when the infection occurred after four doses of mRNA-1273 vaccine [23]. Similarly, a study from Japan reported only marginal increases in anti-S Ab titers in previously infected nursing home residents after receiving the second and third doses of the BNT162b2 vaccine [24]. These findings were consistent with our data. Several studies reported that anti-S antibody levels change over 6 months in participants receiving the vaccination [25,26,27,28]. In the after four-to-six doses group of this study, approximately 30% of HCWs had received only four vaccine doses, and, furthermore, more than 6 months had elapsed since their last vaccination. As a result, the enhancement of anti-S IgG titers due to hybrid immunity following post-vaccination infection may not have been accurately observed. Further investigation is needed to validate this possibility. According to C. H. Fong et al., an enhancement in neutralizing antibody titers against BA.2 and BA.5 due to hybrid immunity was observed in individuals with three or more exposures; however, this enhancement of antibody titers due to hybrid immunity was no longer observed against the ancestral Wuhan strain [29]. The group with three or more exposures was likely infected during the Omicron wave, suggesting that their hybrid immunity had adapted to Omicron variants [29]. In our study, only antibody titers against the ancestral Wuhan strain were measured. Thus, we did not observe an enhancement of antibody titers due to hybrid immunity in the group that received four to six doses (Figure 3A). If antibody titers against Omicron variants had been measured, hybrid immunity could have been detected.

During this study period, the seroprevalence of anti-N Ab remained very low with 0 to 2% from 2020 to 2021), and finally reached 7.9% in June to July 2023) (Table 2). In Japan, where mRNA vaccines are primarily used, anti-N Ab positivity reflects a history of COVID-19. In contrast to our results, the other Japanese HCW reports showed that the seroprevalence of anti-N Ab among HCWs was 6.1% in 2021 at Tokyo Shinagawa Hospital, which is a medium-sized hospital with 300 beds [12]; this is higher than our 1–2% anti-N seroprevalence around the same time. The other study of HCWs working in a larger hospital showed that the seroprevalence of anti-N-Ab in 2021 was 1.6%, 17.7% in 2022, and 54.1% in 2023; this was at Juntendo University Hospital Tokyo, Japan, with 1000 beds (a large-sized hospital) [13]. A study of the general population in a nationwide survey for anti-N seroprevalence among blood donors in Japan indicated the seroprevalence of anti-N IgG was 28.6% in 2022 [30]. Compared to other studies of HCWs and even the general population in Japan, our results showed a lower seroprevalence of anti-N Ab. Therefore, it is assumed that HCWs in a small-sized outpatient clinic resulted in a lower occupational infection risk than those in large-sized hospital due to more stringent infection control measures, as well as the fact that in Japan, private practitioners generally did not treat COVID-19 patients, who were instead admitted to designated hospitals with inpatient facilities.

Some studies have indicated that the factors significantly associated with higher antibody titer include younger age, female, nurse, no smoking history, no hypertension and others after one or two doses of vaccination [8,9,10,11,12,31]. However, our data showed that age, sex, occupation, and smoking history did not influence the anti-S and N IgG Ab titer. In this study group, the correlation between antibodies titer and factors might have diminished. Additionally, while antibody titers typically persist for approximately 3 to 6 months after infection or vaccination for SARS-CoV-2 [25,26], most participants in this study were more than 3 months past their last infection or vaccination. Therefore, the influence of these factors might not have been measurable, and it remains unclear whether the time elapsed or other factors have a detailed causal relationship with antibody titers.

In large hospital settings, nurses were more likely to be seroprevalent and antibody titers than medical doctors [13,32]. Assuming that all healthcare roles had equal access to PPE in outpatient clinics, the higher risk for doctors might be attributed to differences in the frequency, intensity, or duration of patient contact compared to nurses, receptionists, and others. Additionally, outpatient clinics might have been able to implement stricter infection control measures than hospitals.

This study had several limitations. First, blood collection could not be scheduled as planned, such as 1 month and 6 months after vaccination, potentially introducing variability in the timing of antibody level measurements. Considering the rapid decline in SARS-CoV-2 IgG levels within approximately 3 months after vaccination [33], this variability may have significantly impacted the accuracy of our antibody kinetics assessment. Consequently, our findings may be less directly comparable to those of studies that followed stricter sampling schedules. Second, the history of COVID-19 infection was collected solely through interviews, and it was unclear whether the infection was confirmed by RT-PCR testing or rapid diagnostic kits. Third, the methodology of this study was based on antibody titers against the ancestral Wuhan strain and does not account for the potential impact of SARS-CoV-2 variants that appeared over time, such as alpha, delta, or omicron. Fourth, participants whose anti-S IgG Ab titer exceeded the upper limit at any point were treated as having the upper limit titer, which may introduce selection bias. Finally, the high demands of outpatient clinical practice often led to participant dropouts, resulting in a smaller sample size. Furthermore, the analysis was limited as the occupations and medical specialties were restricted to internal medicine and pediatrics.

## 5. Conclusions

In conclusion, we investigated antibody responses to mRNA COVID-19 vaccine among HCWs, examining the effects of baseline characteristics and prior infection. Partial increases in anti-S Ab titer against the ancestral Wuhan strain due to hybrid immunity were observed depending on the timing of the sampling. Multivariate analysis indicated that SARS-CoV-2 infection and vaccination significantly contributed to an increase in anti-S and anti-N IgG Ab titer. It was assumed that HCWs in outpatient clinics might have a lower occupational infection risk than those in large hospital settings, likely due to more stringent infection control measures. The findings of this study provide valuable insights for developing personalized vaccination strategies and future vaccine development.

## Figures and Tables

**Figure 1 vaccines-13-00090-f001:**
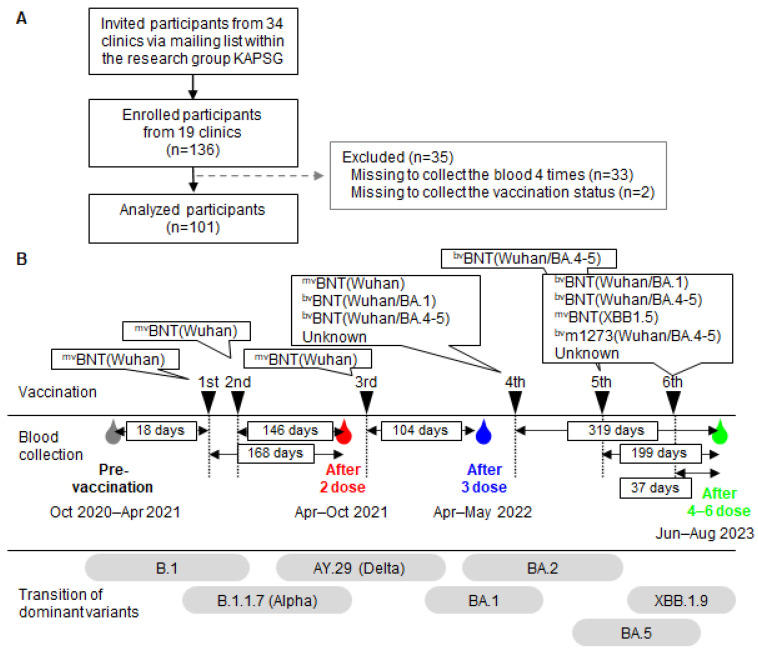
Schematic representations of the study. (**A**) Flow chart for participants’ selection. (**B**) Timeline of vaccination, blood collection, and dominant variants. The days were indicated at median time intervals between vaccination and blood collection. The transition of dominant variants was referenced from Nextstrain (https://nextstrain.org/ncov/gisaid/global/all-time?f_country=Japan, accessed on 24 September 2024). Abbreviations: KAPSG, Kinki Ambulatory Pediatrics Study Group; mv, monovalent; bv, bivalent; BNT, BNT162b2; m1273, mRNA-1273.

**Figure 2 vaccines-13-00090-f002:**
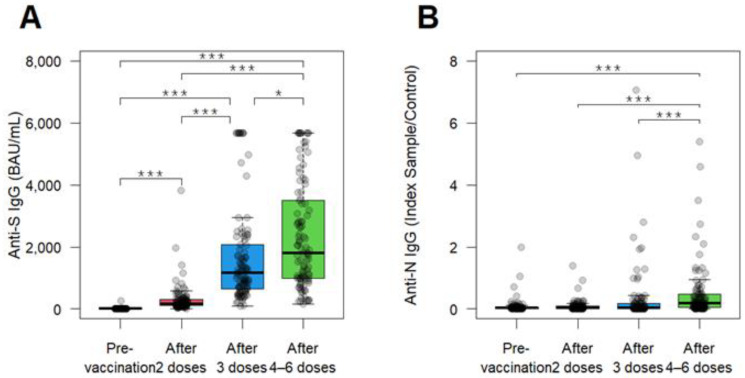
The temporal dynamic of anti-SARS-CoV-2 antibody titers. (**A**,**B**) Anti-S (**A**) and anti-N (**B**) IgG Ab titers in the serum of total analyzed HCWs (n = 101) at 4 time points; pre-vaccination, after 2 doses, after 3 doses, after 4–6 doses. Boxes represent medians, 25th percentiles, and 75th percentiles; whiskers represent maximum and minimum values. Statistical significance was calculated using the Mann–Whitney *U* test (*: *p* < 0.05, ***: *p* < 0.001).

**Figure 3 vaccines-13-00090-f003:**
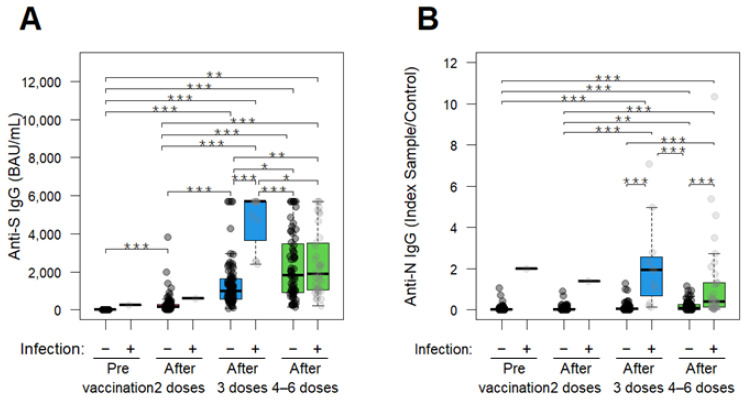
The temporal dynamic of anti-SARS-CoV-2 antibody titers. (**A**,**B**) Anti-S (**A**) and anti-N (**B**) IgG Ab titers in the serum of the total analyzed HCWs (n = 101) at 4 time points; pre-vaccination, after 2 doses, after 3 doses, after 4–6 doses. Boxes represent medians, 25th percentiles, and 75th percentiles; whiskers represent maximum and minimum values. Black and grey dots indicate uninfected (−) and previously infected (+) HCWs, respectively. Statistical significance was calculated using the Mann–Whitney U test (*: *p* < 0.05, **: *p* < 0.01, ***: *p* < 0.001).

**Figure 4 vaccines-13-00090-f004:**
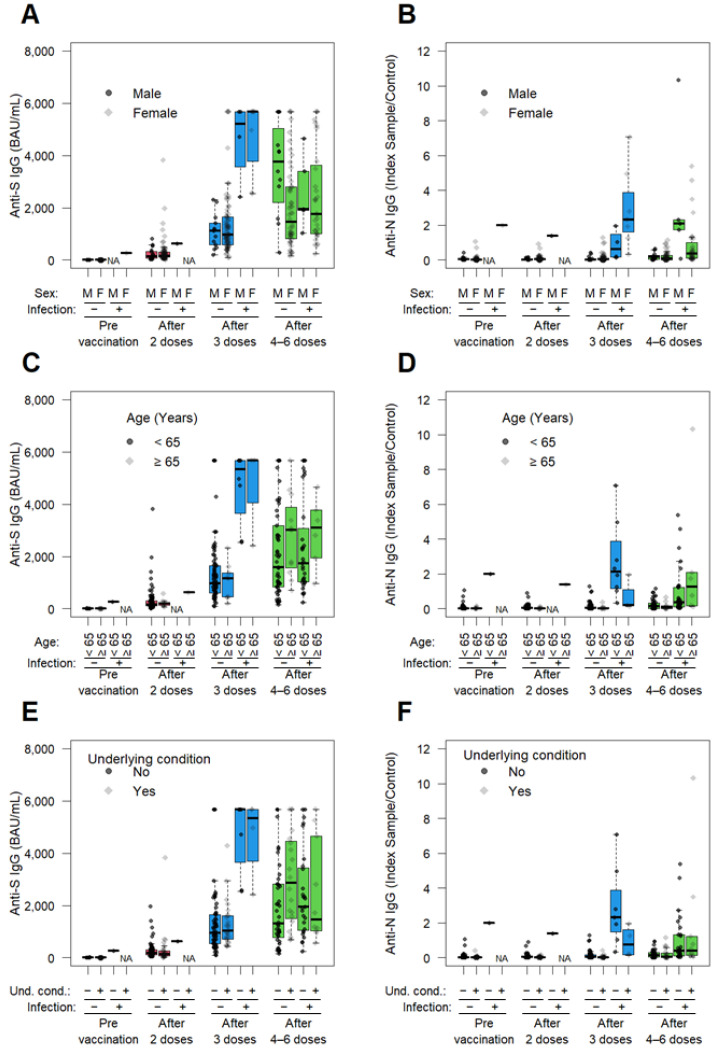
Comparison of anti-SARS-CoV-2 antibodies titer depending on various factors. Anti-S (**A**,**C**,**E**,**G**,**I**) and anti-N (**B**,**D**,**F**,**H**,**J**) IgG Ab titers in the serum of uninfected (−) and infected (+) HCWs were compared across groups. categorized by sex (**A**,**B**), age (**C**,**D**), presence of underlying condition (**E**,**F**), smoking history (**G**,**H**), and occupation (**I**,**J**) at 4 time points; pre-vaccination, after 2 doses, after 3 doses, after 4–6 doses. Boxes represent medians, 25th percentiles, and 75th percentiles; whiskers represent maximum and minimum values. Abbreviations: NA, not applicable; M, male; F, female; D, doctor; N, nurse; R, receptionist; O, others including childminder.

**Table 1 vaccines-13-00090-t001:** Characteristics of healthcare workers in this study.

Characteristics	Total (n = 101)
Age at the time of pre-vaccination (years), median [IQR]	54.0 [46.3, 60.3]
Sex, n (%)	
Male	17 (16.8)
Female	84 (83.2)
Number of vaccinations, n (%)	
1 dose	101 (100)
2 doses	101 (100)
3 doses	100 (99.0)
4 doses	87 (86.1)
5 doses	64 (63.4)
6 doses	40 (39.6)
Type of first vaccine dose, n (%)	
Monovalent BNT162b2 (Wuhan)	101 (100)
Type of second vaccine dose, n (%)	
Monovalent BNT162b2 (Wuhan)	101(100)
Type of third vaccine dose, n (%)	
Monovalent BNT162b2 (Wuhan)	100 (99.0)
Type of fourth vaccine dose, n (%)	
Monovalent BNT162b2 (Wuhan)	43 (42.3)
Bivalent BNT162b2 (Wuhan/BA.1)	8 (7.9)
Bivalent BNT162b2 (Wuhan/BA.4-5)	17 (16.8)
BNT162b2 (Unknown)	18 (17.8)
mRNA-1273 (Unknown)	1 (1.0)
Type of fifth vaccine dose, n (%)	
Bivalent BNT162b2 (Wuhan/BA.4-5)	64 (63.4)
Type of sixth vaccine dose, n (%)	
Bivalent BNT162b2 (Wuhan/BA.1)	2 (2.0)
Bivalent BNT162b2 (Wuhan/BA.4-5)	21 (20.8)
Monovalent BNT162b2 (XBB.1.5)	11 (10.9)
BNT162b2 (Unknown)	1 (1.0)
Bivalent mRNA-1273 (Wuhan/BA.4-5)	1 (1.0)
Unknown	4 (4.0)
Number of infected healthcare workers, n (%)	
Within six months before the first blood sampling	1 (1.0)
Before the second sampling	1 (1.0)
Before the third sampling	11 (10.9)
Before the fourth sampling	37 (36.6)
Occupation, n (%)	
Doctor	20 (19.8)
Nurse	36 (35.6)
Receptionist	40 (39.6)
Childminder	1 (1.0)
Others	4 (4.0)
Underlying condition, n (%)	
Hypertension	15 (14.9)
Heart disease	2 (2.0)
Cancer	2 (2.0)
Others ^a^	11 (10.9)
Smoking ^b^, n (%)	7 (6.9)

^a^ None of the participants had diabetes mellitus. ^b^ Smoking status is defined as current cigarette smoking every day or some days. Abbreviations: IQR, interquartile range.

**Table 2 vaccines-13-00090-t002:** Characteristics of HCWs at each time point of sample collection in this study (n = 84).

Characteristics	Pre-Vaccination	After 1–2 Doses	After 3 Doses	After 4–6 Doses
Anti-S IgG antibody titer (BAU/mL), median [IQR]	0.36 [0.16, 0.54]	163.7 [107.9, 302.0]	1165.5 [656.0, 2073.4]	1808.5 [989.1, 3495.7]
Seroprevalence of anti-S Ab, n (%)	3 (3.0)	100 (99.0)	101 (100)	101 (100)
Anti-N IgG antibody titer (Index Sample/Control), median [IQR]	0.03 [0.02, 0.06]	0.03 [0.02, 0.09]	0.04 [0.02, 0.19]	0.18 [0.05, 0.47]
Seroprevalence of anti-N Ab, n (%)	1 (1.2)	0 (0.0)	6 (5.9)	8 (7.9)

Abbreviations: IQR, interquartile range; S, spike; N, nucleocapsid.

**Table 3 vaccines-13-00090-t003:** Characteristics of uninfected or infected HCWs at each time point of sample collection in this study.

	Pre-Vaccination		After 1–2 Doses		After 3 Doses		After 4–6 Doses	
	Uninfected (n = 100)	Infected (n = 1)	Uninfected (n = 100	Infected (n = 1)	Uninfected (n = 90)	Infected (n = 11)	Uninfected (n = 64)	Infected (n = 37)
Sex, n (%)								
Male	17 (17.0)	NA	16 (16.0)	1 (100)	13 (14.4)	4 (36.4)	12 (18.8)	5 (13.5)
Female	83 (83.0)	1 (100)	84 (84.0)	NA	77 (85.6)	7 (63.6)	52 (81.2)	32 (86.5)
Age (years), median [IQR]	53.88 [46.2, 60.3]	56.58	54.63 [46.7, 60.56]	74.75	53.67 [46.2, 60.7]	60.67 [56.8, 64.5]	56.33 [47.8, 62.5]	56.42 [50.8, 62.3]
Number of vaccinations, n								
1–2 doses	NA	NA	100	1	90	11	64	37
3 doses	NA	NA	NA	NA	89	11	64	36
4 doses	NA	NA	NA	NA	NA	NA	53	34
5 doses	NA	NA	NA	NA	NA	NA	47	17
6 doses	NA	NA	NA	NA	NA	NA	21	8
Anti-S Ab titer (BAU/mL), median [IQR]	0.36 [0.16, 0.53]	270.0	162.4 [107.7, 300.2]	617.5	987.27 [580.1, 1623.3]	5680.00 [3656.7, 5680.0]	1807.30 [918.6, 3433.0]	1899.89 [1040.8, 3495.7]
Anti-N Ab titer (Index Sample/Control), median [IQR]	0.03 [0.02, 0.06]	1.99	0.03 [0.02, 0.08]	1.39	0.04 [0.02, 0.10]	1.93 [0.69, 2.57]	0.10 [0.03, 0.27]	0.39 [0.13, 1.33]

Abbreviations: IQR, interquartile range; NA, not available; Anti-S Ab, anti-SARS-CoV-2 spike antibody; Anti-N Ab, anti-SARS-CoV-2 nucleocapsid antibody.

**Table 4 vaccines-13-00090-t004:** Multivariate regression model to predict anti-S IgG Ab titers after vaccination.

Characteristics	*β* Coefficient	95% CI	*p*-Value
Sex			
Male	Reference	NA	NA
Female	−0.14	[−0.55, 0.27]	0.506
Age			
<65	Reference	NA	NA
≥65	−0.19	[−0.46, 0.15]	0.277
Underlying condition			
No	Reference	NA	NA
Yes	−0.12	[−0.46, 0.21]	0.470
Occupation			
Doctor	Reference	NA	NA
Nurse	0.30	[−0.18, 0.77]	0.221
Receptionist	0.15	[−0.29, 0.59]	0.506
Childminder and others	−0.16	[−0.74, 0.43]	0.598
Number of vaccinations			
None	Reference	NA	NA
2 doses	5.55	[4.92, 6.19]	**<0.001**
3 doses	7.05	[6.42, 7.69]	**<0.001**
4–6 doses	7.40	[6.75, 8.05]	**<0.001**
Previous SARS-CoV-2 infection			
No	Reference	NA	NA
Yes	1.83	[0.71, 2.95]	**<0.01**

Notes: The variables retained in the multivariate regression model to explain anti-SARS-CoV-2 spike IgG antibody titers. Abbreviations: CI, confidence interval; NA, not available. Bold font is used to highlight statistically significant values (*p* < 0.01).

**Table 5 vaccines-13-00090-t005:** Multivariate regression model to predict anti-N IgG Ab titers after vaccination.

Characteristics	*β* Coefficient	95% CI	*p*-Value
Sex			
Male	Reference	NA	NA
Female	−0.38	[−0.95, 0.19]	0.188
Age			
<65	Reference	NA	NA
≥65	−0.30	[−0.70, 0.10]	0.140
Underlying condition			
No	Reference	NA	NA
Yes	−0.38	[−0.68, −0.08]	**<0.05**
Occupation			
Doctor	Reference	NA	NA
Nurse	0.28	[−0.26, 0.26]	0.310
Receptionist	−0.23	[−0.73, 0.81]	0.355
Childminder and others	−0.60	[−1.42, 0.23]	0.158
Number of vaccinations			
None	Reference	NA	NA
2 doses	0.12	[−0.28, 0.52]	0.551
3 doses	0.73	[0.33, 1.15]	**<0.001**
4–6 doses	1.16	[0.70, 1.61]	**<0.001**
Previous SARS-CoV-2 infection			
No	Reference	NA	NA
Yes	2.17	[1.75, 2.59]	**<0.001**

Notes: The variables retained in the multivariate regression model to explain anti-SARS-CoV-2 nucleocapsid antibody titers. Abbreviations: CI, confidence interval; NA, not available. Bold font is used to highlight statistically significant values (*p* < 0.05).

## Data Availability

The research dataset of this study is not publicly available due to privacy restrictions (they are part of an ongoing study). Requests to access the datasets should be directed to Teruhime Otoguro (totoguro@med.niigata-u.ac.jp).

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
