# Peer review of "Antibody Responses to mRNA COVID-19 Vaccine Among Healthcare Workers in Outpatient Clinics in Japan"

_vaccines, 2025, doi:10.3390/vaccines13010090_

Round 1

Reviewer 1 Report (Previous Reviewer 1)

Comments and Suggestions for Authors

The new additions to the manuscript made a big difference. The quality of the paper had improved, and all my questions were addressed. No more comments.

Author Response

Reviewer 1

Comment: The new additions to the manuscript made a big difference. The quality of the paper had improved, and all my questions were addressed. No more comments.

Response: Thank you very much for your positive feedback and for acknowledging the improvements made to our manuscript. We are delighted to hear that all your questions have been addressed satisfactorily.

We truly appreciate your time and effort in reviewing our work, as your insights have significantly enhanced the quality of our paper.

Reviewer 2 Report (Previous Reviewer 3)

Comments and Suggestions for Authors

This revised manuscript examines the humoral immune response to COVID-19 among healthcare workers, employing regression analyses to explore the relationships between immunological outcomes and various contributing factors. By moving beyond a simple observation of immune response levels, the study provides valuable insights into the determinants of these responses. However, the limited sample size poses a challenge, potentially restricting the detection of certain significant associations and representing a limitation of this otherwise well-conducted study. Most concerns raised in previous reviews have been addressed, and the limitations are appropriately acknowledged.

Comments.

1. Diabetes mellitus as an underlying condition:
Was diabetes mellitus included as one of the underlying conditions in the participants?

Diabetes is a known risk factor for severe COVID-19 infection, and its inclusion in the dataset would be valuable.

If this data is available, consider incorporating it into the table of baseline characteristics and including it in the regression analyses to examine its potential association with immunologic outcomes.

2. Visualisation in Figures:
The current plots could benefit from enhancements in visualisation.

Consider using coloured elements with slight transparency to make the plots more visually appealing and easier to interpret, especially when data points overlap.

Author Response

Reviewer 2

This revised manuscript examines the humoral immune response to COVID-19 among healthcare workers, employing regression analyses to explore the relationships between immunological outcomes and various contributing factors. By moving beyond a simple observation of immune response levels, the study provides valuable insights into the determinants of these responses. However, the limited sample size poses a challenge, potentially restricting the detection of certain significant associations and representing a limitation of this otherwise well-conducted study. Most concerns raised in previous reviews have been addressed, and the limitations are appropriately acknowledged.

Comments.

  1. Diabetes mellitus as an underlying condition:
    Was diabetes mellitus included as one of the underlying conditions in the participants?

Diabetes is a known risk factor for severe COVID-19 infection, and its inclusion in the dataset would be valuable.

If this data is available, consider incorporating it into the table of baseline characteristics and including it in the regression analyses to examine its potential association with immunologic outcomes.

Response: Thank you for your insightful comment regarding the inclusion of diabetes mellitus as an underlying condition. In our study, none of the participants were diagnosed with diabetes mellitus. Therefore, this variable was not included in the baseline characteristics or the regression analyses.

  • We have added a footnote to Table 1 to clarify this.

  1. Visualisation in Figures:
    The current plots could benefit from enhancements in visualisation.

Consider using coloured elements with slight transparency to make the plots more visually appealing and easier to interpret, especially when data points overlap.

Response: Thank you for your valuable suggestion regarding the visualization of the plots. In response to your comment, we have revised Figures 2 through 4 to enhance their visual clarity and interpretability. Specifically, we have incorporated colored elements with slight transparency to reduce visual clutter and better distinguish overlapping data points.

We hope these enhancements address your concerns and improve the overall readability of the figures.

This manuscript is a resubmission of an earlier submission. The following is a list of the peer review reports and author responses from that submission.

Round 1

Reviewer 1 Report

Comments and Suggestions for Authors

The authors conducted a study of antibody titers to SARS-CoV-2 vaccines in health care workers from outpatient clinics in Japan. Unfortunately, the study contains a serious methodological flaw and requires a complete revision of the results. The Pfizer-BioNTech and Moderna vaccines produce ONLY anti-S antibodies. It is obvious that antibodies against the N protein are only formed as a result of previous SARS-CoV-2 infection. However, in Figure 3B, there are individuals in the group of uninfected patients who are seropositive for the N protein. These patients had COVID-19 asymptomatically, so they should be excluded from the uninfected group and included in the infected group. Therefore, the groups formed should be revised and all results recalculated.

Lanes 290-292. There were no statistical differences between HCWs with and without previous SARS-CoV-2 infection. Therefore, the conclusion "Our results suggest that hybrid immunity increased anti-S IgG Ab in vaccinated HCWs with a prior SARS-CoV-2 infection" is incorrect. This should be changed throughout the manuscript.

Specific comments:

The title of the manuscript does not clearly define " their predictors". Please change or delete this term.

Lines 45-46. Please clarify which vaccines were used in Japan during the specified period (trade name and type of vaccine). The characteristics of the vaccines used, including the characteristics of the immune response to these vaccines, should be provided here.

The quality of Figure 4 is poor, please correct it.

Reviewer 2 Report

Comments and Suggestions for Authors

Estimated Authors,

I've read with great interest the present paper entitled "Antibody responses to mRNA COVID-19 vaccine and their predictors among healthcare workers in outpatient clinics in Japan".

In this study, Otoguro et al. provide a seroprevalence and serological assessment report from HCWs from various prefectures from Japan since the onset of SARS-CoV-2 vaccination campaigns. The present paper suggest that factors such as age, gender, occupation have a limited  effect on the level of anti-S IgG (with repeated doses as the main and sole explanatory variables significantly associated with Ig levels), while anti-N IgG antibodies (i.e. those only elicited by natural infection) were associated with working as a nurse, receiving 3 or 4-6 doses of vaccine, and having been previously vaccinated.

In fact, several explanations could be advocated, including the high risk for natural infection of nurses due to their occupational duties compared to medical professionals.

Albeit interesting, the present paper is affected by several shortcomings, some of them could be solved by Authors through text editing. More precisely:

1) the number of sampled professionals is very small (< 150 HCWs). moreover, these HCWs are occupationally heterogeneous; authors are welcome to provide a more accurate appraisal of the reduced sample size as a potential issue for their study;

2) Authors should explain in further details how the sample was identified and finally recruited. How was the total number of recruited HCWs determined? was it a convenience sample? Was a preliminary power analysis performed?

3) How were HCWs recruited? by invitation by means of their institution?

4) Limits acknowledged by Authors in Limits section include some significant issues:

a) "First, blood collection could not be scheduled as planned, such as one month and six months after vaccination". Authors should discuss far more extensively this issue through available evidence from other studies, as there is a substantial consensus about the rapid decrease of Ig levels after SARS-CoV-2 vaccination

b) "Fourth, participants whose anti-S IgG Ab titer exceeded the upper limit at any point were excluded, which may introduce selection bias" Authors must explain how this specific topic was addressed in logistic regression.

Reviewer 3 Report

Comments and Suggestions for Authors

This study delineates the humoral immune response to COVID-19 among healthcare workers, utilising regression analyses to examine the relationship between immunologic outcomes and various contributing factors. This approach provides valuable insights beyond merely observing immune response levels. However, the limited sample size may restrict the detection of certain significant associations, representing a limitation of this study.

Major Concerns.

1. Lines 93-94: Please specify the type of Omicron subvariants used in the vaccines in this study, for instance, Original/BA.1, Original/BA.2, or Original/BA.4/5.

Also, when referring to the “monovalent” vaccine, does this term indicate the first-generation COVID-19 vaccine (ancestral/wild-type/original strain), or is it referring to the XBB.1.5 monovalent formulation? 
Please clarify.

2. Specialty Classification: Were medical doctors in the study categorised by specialty? 

If so, did the analysis reveal any additional insights based on specialty? Certain specialties, such as pulmonology, ENT, and infectious disease, may be associated with higher exposure risks.

Comments.

1. Lines 105-111: Please rewrite the description of the serologic assessment tools for clarity.

The assessment in this study included the analyser [Architect i2000SR (Abbott, Chicago, IL)], IgG anti-N [SARS-CoV-2 IgG (Abbott, Chicago, IL)], and IgG anti-RBD [SARS-CoV-2 IgG II Quant or AdviseDx SARS-CoV-2 IgG II (Abbott, Chicago, IL)].
Please see the reagent's leaflets.

Additional resources:

- https://www.corelaboratory.abbott/int/en/offerings/brands/architect/architect-i2000SR.html

- https://www.corelaboratory.abbott/int/en/offerings/segments/infectious-disease/sars-cov-2.html

- https://www.fda.gov/media/137383/download

- https://www.fda.gov/media/146372/download

2. IgG anti-RBD: This test is available in BAU/mL (Binding Antibody Units per mL), which is a standardised unit for inter-assay comparison.
I suggest using BAU/mL instead of AU/mL, and including a conversion factor in the text.
This test’s conversion factor from AU/mL to BAU/mL is 0.142:

BAU/mL = AU/mL x 0.142

Reference:

- https://pmc.ncbi.nlm.nih.gov/articles/PMC8847080

3. Figures in colour: Since this journal is an online publication, consider using colour figures instead of greyscale to enhance clarity.

4. Line 352 (Handling ceiling effect in some samples): The exclusion of participants whose anti-S IgG titres exceeded the instrument’s upper limit is noted.
If there are leftover serum/plasma samples, it may be helpful to perform a dilution (e.g., with NSS or PBS, or following manufacturer guidelines) and remeasure, then calculate the actual titre by applying the dilution factor.

Typos/Errors:

1. Figure 1: Please check for singular/plural consistency. Errors noted include “2 dose,” “3 dose,” and “4-6 dose” rather than the correct “2 doses,” “3 doses,” and “4-6 doses.”

2. Lines 179-180: The anti-N measurement unit is currently reported as “index S/N.” It appears this should be “index S/C.” 

Please revise accordingly.